# Benefits of Linear Conditioning with Metadata for Image Segmentation

**Andreanne Lemay** [1,2]                                     ANDREANNE.LEMAY@POLYMTL.CA
[1] *NeuroPoly Lab, Institute of Biomedical Engineering, Polytechnique Montreal, Canada*
[2] *Mila, Quebec AI Institute, Canada*
**Charley Gros**[1,2]                                          CHARLEY.GROS@GMAIL.COM
**Olivier Vincent**[1,2]                                       OVINCENT.POLY@GMAIL.COM
**Yaou Liu**[3]                                                YAOULIU80@163.COM
[3] *Beijing Tiantan Hospital, Capital Medical University, China*
**Joseph Paul Cohen**[*2,4]                                    JOSEPH@JOSEPHPCOHEN.COM
[4] *Stanford University Center for Artificial Intelligence in Medicine & Imaging*
**Julien Cohen-Adad**[*1,2,5]                                  JCOHEN@POLYMTL.CA
[5] *Functional Neuroimaging Unit, CRIUGM, University of Montreal, Montreal, Canada*

## Abstract

Medical images are often accompanied by metadata describing the image (vendor, acquisition parameters) and the patient (disease type or severity, demographics, genomics). This metadata is usually disregarded by image segmentation methods. In this work, we adapt a linear conditioning method called FiLM (**F**eature-w**i**se **L**inear **M**odulation) for image segmentation tasks. This FiLM adaptation enables integrating metadata into segmentation models for better performance. We observed an average Dice score increase of 5.1% on spinal cord tumor segmentation when incorporating the tumor type with FiLM. The metadata modulates the segmentation process through low-cost affine transformations applied on feature maps which can be included in any neural network's architecture. Additionally, we assess the relevance of segmentation FiLM layers for tackling common challenges in medical imaging: multi-class training with missing segmentations, model adaptation to multiple tasks, and training with a limited or unbalanced number of annotated data. Our results demonstrated the following benefits of FiLM for segmentation: FiLMed U-Net was robust to missing labels and reached higher Dice scores with few labels (up to 16.7%) compared to single-task U-Net. The code is open-source and available at www.ivadomed.org.
**Keywords:** Deep learning, linear conditioning, segmentation, metadata, task adaptation.

## 1. Introduction

Segmentation tasks in the medical domain are often associated with metadata: medical condition of the patients, demographic specifications, acquisition center, acquisition parameters, etc. Depending on which structure is segmented, these metadata can help deep learning models improve their performance, however, metadata is usually overlooked. In this work, we improve segmentation models using recent advances in visual question answering called FiLM (Perez et al., 2018; de Vries et al., 2017) (**F**eature-w**i**se **L**inear **M**odulation). Using FiLM to condition a segmentation model enables the integration of prior metadata into neural networks through linear modulation layers. For instance, knowledge of the

---

* Contributed equally

tumor type could provide useful information to the model. (Rebsamen et al., 2019) demonstrated that by stratifying the learning by brain tumor type, high-grade glioma, or low-grade glioma, segmentation could be improved. With FiLM, the tumor type information can be included without requiring multiple models as done in (Rebsamen et al., 2019). The input metadata generates feature-specific affine coefficients learned during training, enabling the model to modulate the segmentation output to improve its performance.

The metadata could also be exploited for task adaptation. When training a multi-class segmentation model, each class needs to be annotated on every image, as missing labels will hamper the learning (Zhou et al., 2019). Label availability often represents a bottleneck in deep learning (Minaee et al., 2020). Segmentation is costly in terms of time, money, and logistics (Bhalgat et al., 2018). For instance, chest CT scans contain hundreds of 2D scans (up to 861 axial slices in the dataset used for this work) depending on the resolution. As a reference, Google sets the price of image segmentation to 870 USD for 1000 images [1], which totals 435 USD for a single subject with 500 axial slices. For medical segmentation requiring expert knowledge (e.g., tumor segmentation), this price could be higher considering the hourly wage of a radiologist. As for the time, (Ciga and Martel, 2021) reports that it takes between 15 minutes and two hours depending on the size and resolution to segment a single image of lymph nodes for breast cancer. An approach dealing with missing modalities and requiring fewer labels can reduce the monetary and time-related costs.

We hypothesize that conditioning the model based on the organ to be segmented (e.g., "kidney", "liver") will make it robust to missing segmentations. A multi-class model could then be trained on data from multiple datasets with a single class annotated in each. Since the different tasks share weights, fewer labels are required for a given class as the model can learn from the other tasks. This enables the model to easily adapt a single segmentation model to several tasks requiring only a small amount of annotations for novel tasks.

## 1.1. Prior work

Conditional linear modulation was introduced in many deep learning fields: visual reasoning (Perez et al., 2018; de Vries et al., 2017), style transfer (Dumoulin et al., 2017), speech recognition (Kim et al., 2017), domain adaptation (Li et al., 2018), few-shot learning (Oreshkin et al., 2018), to name a few. In the medical image field, FiLM was leveraged for learning when limited or no annotation is available for one modality (Chartsias et al., 2020). Image reconstruction was performed with FiLM to enable self-supervised learning of the anatomical and modality factors of an image. Modality factors were passed through FiLM to modulate anatomical factors generating a reconstructed image of a given modality. While in (Chartsias et al., 2020) information extracted from the image is used for modulation, in this work, we want to assess the impact of integrating metadata that is not directly encoded in the image.

The adaptation of FiLM (i.e., linear conditioning) for segmentation was experimented on cardiovascular magnetic resonance modulated by the distribution of class labels (Jacenków et al., 2019), on ACDC with modulation on spatio-temporal information (Jacenków et al., 2020) and on multiple sclerosis lesions with a FiLMed U-Net conditioned on the modality (T2-weighted or T2star-weighted) (Vincent et al., 2020). (Jacenków et al., 2019) had con-

---

1. https://cloud.google.com/ai-platform/data-labeling/pricing

sistent improvement by including the prior information on an encoder-decoder architecture but mitigated results on the U-Net architecture. Results from (Vincent et al., 2020) were inconclusive regarding the performance of FiLM compared to a regular U-Net. A possible explanation for this lack of improvement is that the modality-related features might already be encoded in the regular U-Net, therefore the metadata added to FiLM is not informative enough and thus does not translate to an increase in segmentation performance. In light of these results, in the present work, we generalized the modified-FiLM implementation to be able to modulate a model by inputting any type of discrete metadata data.

## 1.2. Contribution

The key contributions of this work are: **(i)** We introduce an adaptation of linear conditioning (Perez et al., 2018) based on metadata for segmentation tasks using the U-Net architecture. **(ii)** We demonstrate that including metadata can contribute to the model's performance. As a proof of concept, we input the spinal cord tumor type (astrocytoma, ependymoma, hemangioblastoma), which is often associated with its size, composition, and anatomical location. The tumor type knowledge led to an average Dice score improvement of 5.1%. **(iii)** We show that robust learning with missing annotations can be achieved with FiLM. Moreover, we illustrate that linear modulation enables task adaptation with fewer labeled data when jointly trained on multiple tasks. A Dice score improvement of up to 16.7% was observed when using our approach with a limited number of annotations compared to a single class U-Net.

## 2. Methods

### 2.1. Architecture and Implementation

The core architecture is based on the 2D U-Net (Ronneberger et al., 2015) (Figure 1). The model has two inputs: the image and the one-hot encoded metadata (i.e., prior knowledge). FiLM layers and generator are responsible for conditioning the neural network with the given metadata. Two parameters, $\gamma_{(i)}$ and $\beta_{(i)}$, are required to linearly modulate the inputs of the $i^{th}$ FiLM layer. The metadata is passed through a multi-layer perceptron (i.e., FiLM generator) with two hidden layers (64 and 16 neurons). The FiLM generator outputs one value of $\gamma$ and $\beta$ for each filter (i.e., feature extractor) which are respectively multiplied and added by the FiLM layers to each convolutional feature map. The computational cost of FiLM is low and independent of the image resolution. The weights from the generator are shared for a more efficient learning (Perez et al., 2018). Since the input of the FiLM generator is the same, the same features should be extracted from the metadata. The values are constrained between 0 and 1 due to the sigmoid activation. Preliminary experiments favored sigmoid over ReLU or tanh activation function for the FiLM parameters. $\gamma_{(i)}$ values near 0 silence some features, while $\gamma_{(i)}$ values near 1 output the key features. Since the linear modulation is computationally inexpensive, FiLM layers were placed after each convolutional unit to ensure the metadata is properly used by the network. The code is open-source and available in the ivadomed toolbox (Gros et al., 2021).

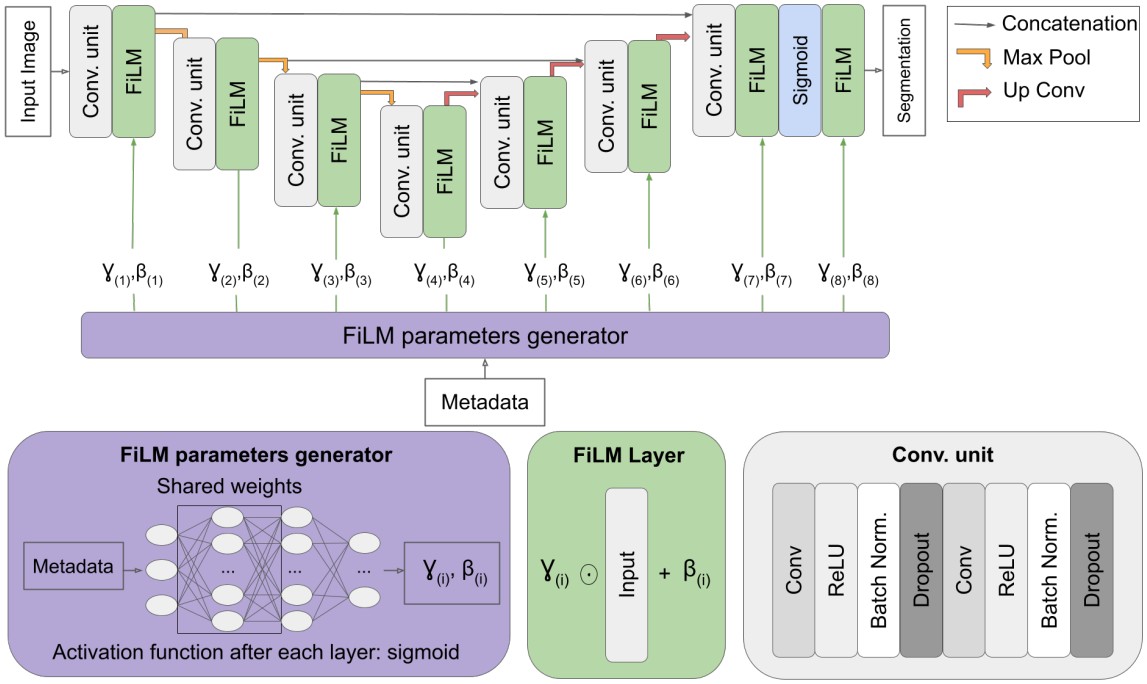

Figure 1: FiLMed U-Net architecture of depth 3. Depth describes the number of maximum pooling or up convolutions in the U-Net. $\gamma$ and $\beta$ values are generated using a multi-layer perceptron with shared weights across FiLM layers. $\gamma$ and $\beta$ have the same shape as the input. An element-wise multiplication is applied between the input and $\gamma$ while the $\beta$ is added.

## 2.2. Experiment 1: Segmentation using relevant metadata

This experiment assessed the relevance of including metadata during the training.

### 2.2.1. DATASET: SPINAL CORD TUMOR

We used a spinal cord tumor segmentation dataset (Lemay et al., 2021). The dataset included 343 MRI scans, where each image was associated with one of the following tumor types: astrocytoma (101), ependymoma (122), or hemangioblastoma (120). The tumor type can be informative for the model since each type has particular characteristics, e.g., size, location, contrast intensity patterns, tissue constitution, (Kim et al., 2014; Baleriaux, 1999). Two modalities, Gadolinium-enhanced T1-weighted (Gd-e T1w) and T2-weighted (T2w), are required to properly segment each component of the tumor: tumor core, edema, and liquid-filled cavity. Here, for simplicity, only the tumor core labels were used.

### 2.2.2. TRAINING SCHEME

The first scenario used the FiLM architecture without any input metadata, while the second scenario included the tumor type as metadata. To simulate the absence of metadata, the same input vector was passed through FiLM, hence no informative data is seen by the model. The same architecture was used in both scenarios in order to isolate the specific effect of the input metadata. Preliminary experiments gave similar results when using a regular U-Net architecture without the FiLM layers or a FiLMed U-Net with always the same input. A 320x256 sagittal image of resolution 1mmx1mm associated with the tumor type constituted

one training sample. The dataset was split per patient with the following proportions: 60% training, 20% validation, 20% testing. To compare the overall segmentation performance, 10 models were trained with different random splits.

### 2.3. Experiment 2: FiLM for multiple tasks

Here, the ability of FiLM to modulate the network to adapt to different segmentation tasks was assessed. The FiLMed model was presented with labels from three classes that are all included in the scan, but only one segmentation was given at the time. The class of the presented segmentation was input into the network to teach the model to properly segment each class. A similar experiment was performed with few segmentations and unbalanced datasets.

#### 2.3.1. Dataset: Spleen, kidneys, and liver

The organs selected for this task were the spleen, kidneys, and liver. The datasets were collected from two different sources: Medical Segmentation Decathlon (Simpson et al., 2019) for spleen and liver scans, and KiTS19 (Heller et al., 2019) for kidney scans. Liver and kidney scans had tumor labeling which was ignored for the current experiments: organ and tumor annotations were merged as a single segmentation. Due to the large size of the kidney and liver datasets, subdatasets were extracted. Since the spleen dataset contained 41 scans with associated ground truths, only the first 41 kidney and liver scans were retained.

#### 2.3.2. Training scheme

First, the FiLMed U-Net was trained on the spleen, kidney, and liver images with the whole dataset (41 images for each). A training example was a 2D axial slice of 512x512 pixels paired with the available label (kidney, spleen, or liver). The dataset was split per patient with the following proportions: 60% training, 20% validation, 20% testing.

Second, the performance on small and unbalanced datasets was assessed with an independent sub-experiment: FiLMed U-Net was trained on subdatasets of the spleen and kidney datasets. For simplicity, only two classes were used. The experimental design of this sub-experiment is presented in appendix A. The subdatatsets were randomly chosen with a size of 2, 4, 6, 8, and 12 for one class and 12 subjects of the other class (i.e., a total of 10 models: {2, 4, 6, 8, 12} spleens with 12 kidneys each and {2, 4, 6, 8, 12} kidneys with 12 spleens each). The size of the dataset included all the subjects for training and validation. The models were tested on 25 subjects of the class with the least subject. For a model trained on 2 kidney subjects and 12 spleen subjects, the model would be tested on 25 kidney subjects not included in the training or validation set. During the training process, the data was sampled to expose each class evenly to the model even when the number of subjects is unbalanced. All the trainings were repeated 10 times with varying random splits (100 trainings).

Regular 2D U-Nets trained on only one class at the time, spleen, kidney, or liver were trained following the same training, validation, and test splits for comparison.

### 2.4. Training parameters

The tumor types or organ labels were evenly separated into three groups, training, validation, and testing groups, and the data were sampled with a batch size of 8. The FiLMed

U-Nets of depth 4 for the spinal cord tumor and 5 for the chest CT were trained with a Dice loss function until the validation loss plateaued for 50 epochs (early stopping with $\epsilon = 0.001$). The depth was chosen according to the size of the input images. The initial learning rate was 0.001 and was modulated according to a cosine annealing learning rate.

### 2.5. Evaluation

The Dice score was selected to compare the performance of each approach. All FiLMed approaches were compared with the conventional approach: training without informative metadata for spinal cord tumors and on a regular U-Net for the multi-organ segmentation tasks. To assess the statistical differences between groups, a one-sided Wilcoxon signed-rank test with a p-value $< 5\%$ was considered to be a significant difference.

## 3. Results

### 3.1. Experiment 1: Segmentation using relevant metadata

Prior knowledge of the tumor type led to a significant Dice score improvement between the regular U-Net and the FiLMed U-Net: 10.5% for the hemangioblastomas (p-value=0.006), 4.5% for the astrocytomas (p-value=0.003), and 5.1% for all tumors combined (p-value=0.003) (Table 1). Astrocytomas and hemangioblastomas showed the highest Dice score gain when the model was informed with the tumor type. Astrocytomas are typically large, have ill-defined boundaries, and present heterogeneous, moderate, or partial enhancement in the Gd-e T1w contrast (Baleriaux, 1999). Conversely, hemangioblastomas are usually associated with a small tumor core (Baleriaux, 1999) intensely enhanced on Gd-e T1w (Baker et al., 2000). These distinctive characteristics can be learned by the model to perform a more informed segmentation (see appendix B to visualize segmentation differences).

Table 1: Spinal cord tumor core segmentation performance for regular and FiLMed U-Net (mean ± STD % for 10 random splits). The FiLMed U-Net was trained with the tumor type as input. ** p-value < 0.05 for one-sided Wilcoxon signed-rank test.

| | Dice score [%] | |
| --- | --- | --- |
| **Tumor type** | **No prior info.** | **Prior info.** |
| Astrocytoma | $53.3 \pm 4.8$ | $\mathbf{57.8 \pm 4.9}$ ** |
| Ependymoma | $57.2 \pm 3.2$ | $\mathbf{57.7 \pm 2.4}$ |
| Hemangioblastoma | $51.2 \pm 4.0$ | $\mathbf{61.7 \pm 3.7}$ ** |
| **All** | $54.0 \pm 2.2$ | $\mathbf{59.1 \pm 2.3}$ ** |

### 3.2. Experiment 2: FiLM for multiple tasks

Table 2 shows that the FiLMed multi-class model trained with missing labels (i.e., only one organ labeled per scan) was able to reach equivalent performance to single-class U-Nets (i.e., one model per class) trained without missing annotations. As a reference, a multi-class 2D U-net without FiLM was trained with the same dataset containing missing labels. Poor performance was reached with an average Dice score of $41.7 \pm 16.0$ for all classes combined:

Table 2: Multiple-organ segmentation Dice score with multi-class, single-class and FiLMed U-Nets (mean ± STD %). The FiLMed U-Net was trained on spleen, kidney, and liver while regular U-Nets were trained on each class independently. A one-sided Wilcoxon signed-rank test was performed on columns 2 (2D U-Net) and 3 (FiLMed U-Net): no statistical difference was observed.

| | Our experiments | | | Literature |
|---|---|---|---|---|
| **Task** | **Multi-class 2D U-Net** | **Single-class 2D U-Net** | **Multi-class FiLMed U-Net** | **2D U-Net (On whole challenge dataset)** |
| Liver | $50.3 \pm 18.3$ | $95.1 \pm 1.4$ | $94.1 \pm 1.6$ | $94.37 \pm N/A$ (Isensee et al., 2018) |
| Spleen | $35.6 \pm 14.2$ | $91.7 \pm 6.3$ | $92.2 \pm 5.3$ | $94.2 \pm N/A$ (Isensee et al., 2019) |
| Kidney | $39.2 \pm 13.1$ | $90.4 \pm 9.3$ | $90.7 \pm 8.1$ | $93.0 \pm 1.2$ (Ahmed, 2020) |

only partial segmentation of each organ was performed by the model. This result illustrates the hindered learning caused by the missing annotations. Inputting the class label through FiLM layers allowed the model to properly train with missing segmentations enabling the option to have a single model adapted to multiple tasks even when all annotations are not available. For comparison, the Dice scores reached by other studies on the whole challenge datasets, 61 spleens, 300 kidneys, and 201 livers, with 2D U-Nets was included. While being trained on less data (41 images per dataset), our 2D FiLMed U-Net reached Dice scores comparable with these published studies (see Table 2).

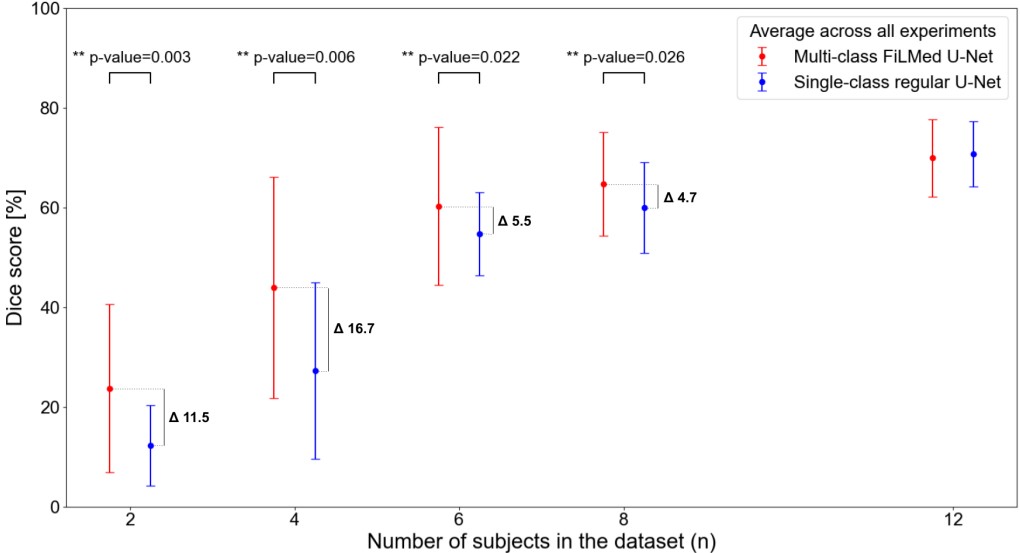

Figure 2: Spleen and kidney segmentation Dice scores for small and unbalanced datasets. The number of subjects combines training and validation subjects. Dice scores for all experiments on the test set (25 subjects) were averaged across the number of subjects and aggregated according to the approach, FiLMed (red) or regular U-Net (blue). The error bars show the standard deviation. $\Delta$ indicates the difference of mean Dice scores between the two approaches. The data totals 10 models trained on different random splits. ** p-value $< 5\%$ with one-sided Wilcoxon signed-rank test.

Figure 2 demonstrates the ability of FiLMed U-Net to be trained on small or unbalanced datasets. With the same amount of labels for a given class, FiLMed models reached superior Dice scores for datasets of size 2, 4, 6, and 8 compared with the regular U-Nets trained on a single class, 11.5%, 16.7%, 5.5%, and 4.7%, respectively. This suggests that the FiLMed models were able to learn from the images associated with the other task. The more subjects are included in the dataset, the more similar FiLM performances become to regular U-Nets, as seen in Table 2. However, FiLMed models have the advantage of being robust to missing classes.

## 4. Discussion

FiLM provides a flexible, low computational cost option to integrate prior knowledge. In this paper, the type of spinal cord tumor was exploited as a proof of concept, but the possibilities of metadata that can improve the performance of a model are vast. The prior metadata could include domain information (e.g., acquisition center, scanner vendor), anatomical data (e.g., location in the body, pose estimation, disease type or severity), or rater specification (e.g., rater's experience, rater's id). To elaborate on an example, inter-expert variability is an important aspect in medical segmentation (Renard et al., 2020). Integrating this information in the model would enable one to make predictions according to the rater with the most experience or to create a model that can replicate inter-expert predictions (i.e., generating one prediction per expert learned in training).

FiLM is capable of dealing with missing labels by indicating which annotations are presented to the model. Many new medical imaging datasets are available, however, most have limited scopes and annotations. FiLM makes it possible to use data from different sources with only one class annotated to create a multi-class model instead of single-class models trained on each dataset. Without the need for more labels, combining datasets increases the number of examples seen by the model. Since weights are shared between tasks, the model learns from the data of the other tasks as seen in Figure 2. The transfer learning between tasks and the robustness with respect to missing segmentations reduce the number of annotations required.

Since the metadata is one-hot encoded before being introduced into the FiLM generator, discrete prior information is needed. The approach presented works with continuous data (e.g., age, size, MRI acquisition parameters), but it must be discretized into a binned range. Future work should explore methods to best encode different data types. This enhancement would allow the integration of MRI acquisition parameters (e.g., echo-time, flip angle) that might make the model agnostic to the different acquisition sequences.

## 5. Conclusion

The integration of linear conditioning through FiLM for segmentation models enables a flexible option to integrate metadata to enhance the predictions. FiLM also facilitates the training of multi-class models by being robust to missing labels. Future work could focus on the impact of integrating other types of data than the tumor type, increasing the number of metadata used to modulate the network, or evaluating the impact of including prior information on the model's uncertainty.

## Acknowledgments

We thank the contributors of the ivadomed project, Lucas Rouhier, Ainsleigh Hill, Valentine Louis-Lucas, and Christian Perone for fruitful discussions.

Funded by the Canada Research Chair in Quantitative Magnetic Resonance Imaging [950-230815], the Canadian Institute of Health Research [CIHR FDN-143263], the Canada Foundation for Innovation [32454, 34824], the Fonds de Recherche du Québec - Santé [28826], the Fonds de Recherche du Québec - Nature et Technologies [2015-PR-182754], the Natural Sciences and Engineering Research Council of Canada [RGPIN-2019-07244], the Canada First Research Excellence Fund (IVADO and TransMedTech), the Courtois NeuroMod project and the Quebec BioImaging Network. This research is based on work partially supported by the CIFAR AI and COVID-19 Catalyst Grants. A.L. has a fellowship from NSERC, FRQNT, and UNIQUE, C.G. has a fellowship from IVADO [EX-2018-4], O.V. has a fellowship from NSERC, FRQNT, and UNIQUE.

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

## Appendix A. Experimental design of organ segmentation with limited annotations

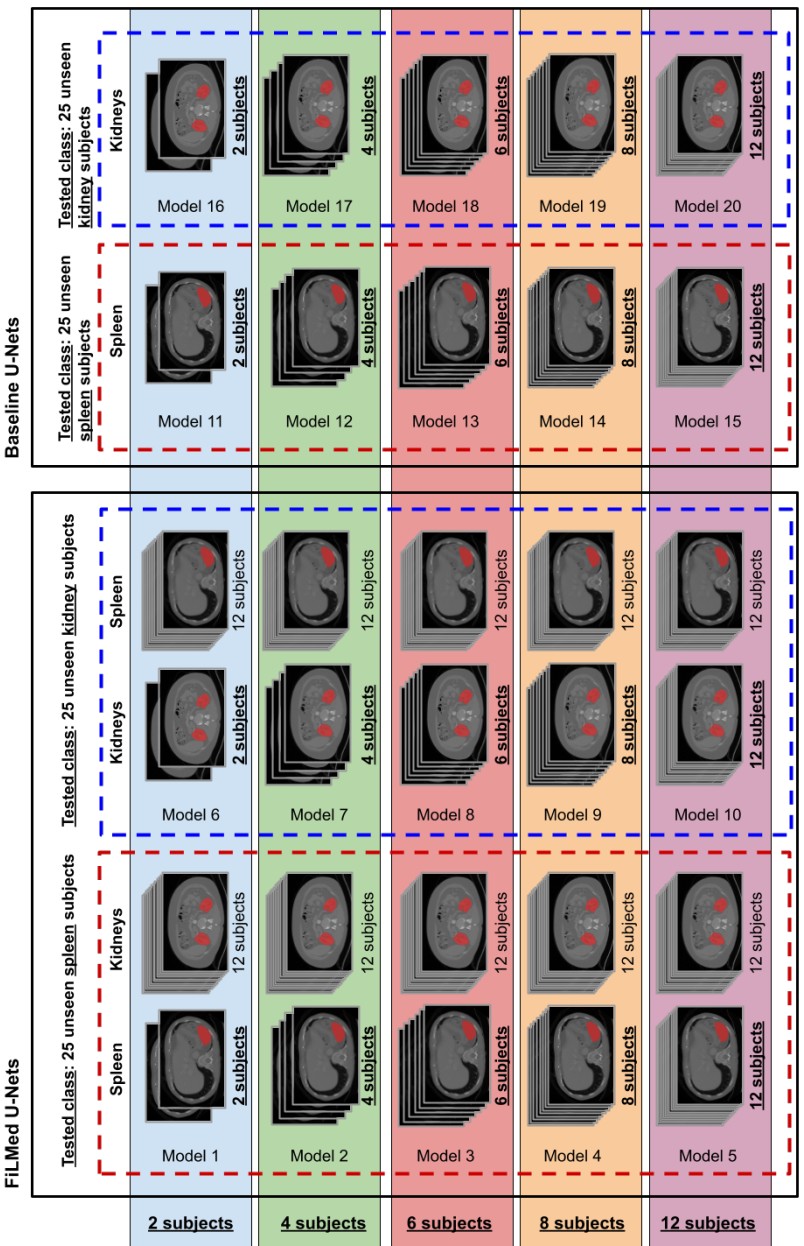

Figure 3: Experimental design of organ segmentation with limited annotations. The images associated to each model represent the training and validation set. This experimental design was used to generate Figure 2.

## Appendix B. Spinal cord tumor segmentation

Figure 4: Tumor segmentation prediction by FiLMed U-Net informed by the tumor type, "With prior", or not informed, "No prior". A1 and A2 presents two subjects with astrocytomas. H1 and H2 presents two subjects with hemangioblastomas. GT: Ground truth.

Astrocytomas are typically large, have ill-defined boundaries, and present heterogeneous, moderate, or partial enhanced in the Gd-e T1w contrast (Baleriaux, 1999). Astrocytomas are usually extensive, expanding from 2 to 19 vertebral bodies in size (Baleriaux, 1999). In both A1 and A2 predictions from the model without prior information, the segmented tumor size was one vertebral body or less and corresponded to the most enhanced tumor signal on the Gd-e T1w (ignoring the rest of the lesion).

In counterpart, hemangioblastomas are usually associated with a small tumor core (Baleriaux, 1999) intensely enhanced on Gd-e T1w (Baker et al., 2000). Figure 4 H1 presents a hemangioblastoma barely apparent in T2w and hidden by the cavity (hyperintense signal). The small hyperintense signal on the Gd-e T1w contrast was overseen by the regular approach. On H2, the model oversegmented the tumor and identified a second tumor on a hypointense signal. The false positive tumor identification does not present an intense Gd-e T1w enhancement which is usually the case for hemangioblastomas. This false positive is not present for the model informed by the tumor type.

To assess the impact of inputting the tumor type, each prediction was modulated by the different tumor types. Table 3 presents the quantitative results for each condition while Figure 5 qualitatively illustrates the impact of changing the tumor type. The highest Dice scores are reached when the input label corresponds to the true label. The modulation

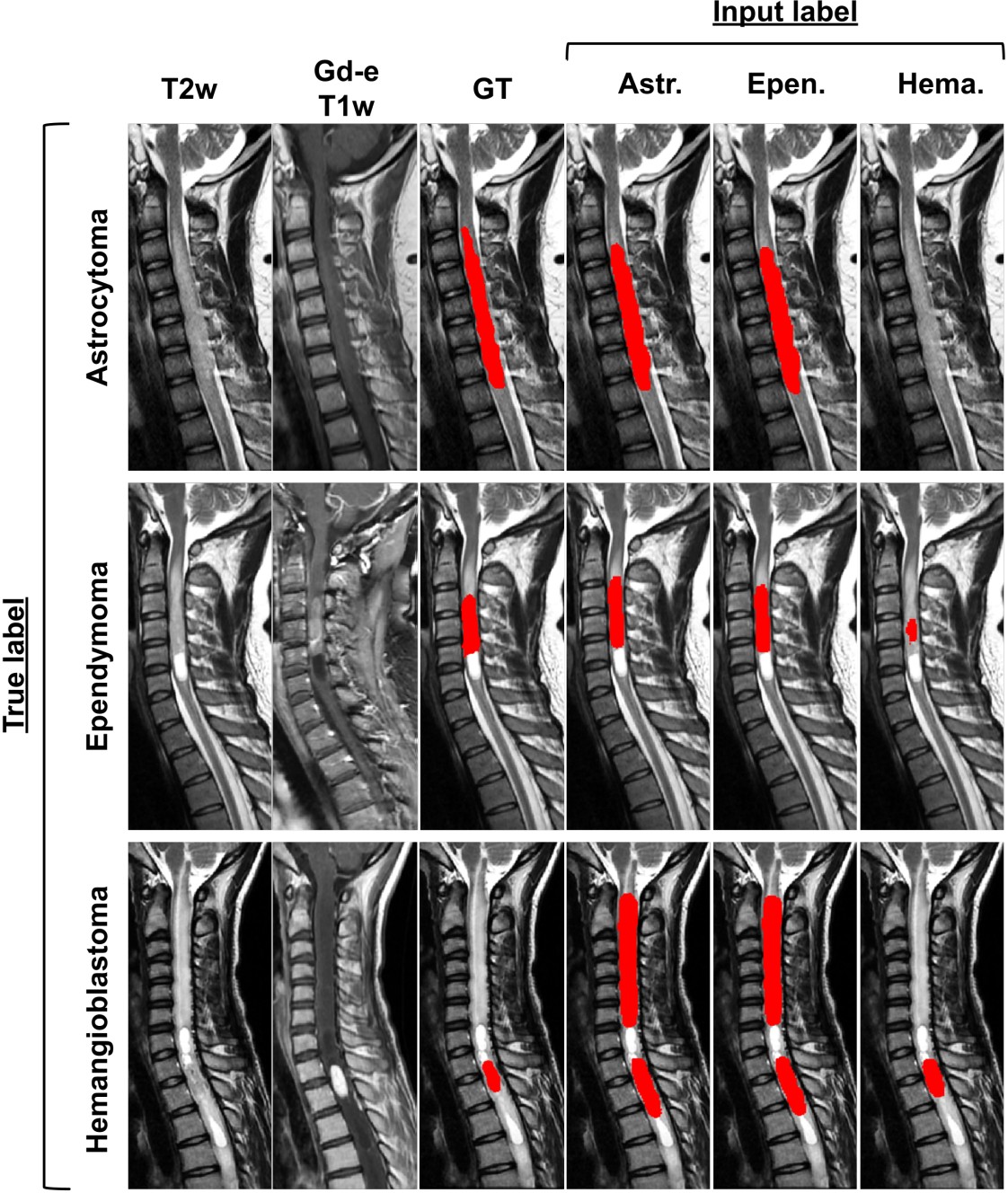

Figure 5: Impact of inputting different tumor types with FiLMed U-Net on the model's segmentation. True label represents the tumor type while input label is the tumor type input into the model through FiLM. Astr.: Astrocytoma, Epen.:Ependymoma, Hema.: Hemangioblastoma.

Table 3: Spinal cord tumor core segmentation Dice scores for FiLMed U-Net with the different tumor types as input (mean $\pm$ STD % for 10 random splits). True label represents the tumor type while input label is the tumor type input into the model through FiLM. ** p-value $< 0.05$ for one-sided Wilcoxon signed-rank test compared to the highest value in each row.

| | Input label | | |
|---|---|---|---|
| **True label** | **Astrocytoma** | **Ependymoma** | **Hemangioblastoma** |
| **Astrocytoma** | **57.9 $\pm$ 4.9** | 57.3 $\pm$ 4.9 | 32.2 $\pm$ 5.1 ** |
| **Ependymoma** | 57.6 $\pm$ 2.6 | **57.7 $\pm$ 2.4** | 35.9 $\pm$ 4.7 ** |
| **Hemangioblastoma** | 41.5 $\pm$ 4.7 ** | 41.8 $\pm$ 6.4 ** | **61.7 $\pm$ 3.7** |

with FiLM successfully encoded knowledge about the tumor types and the predictions are in agreement with known characteristics of the different types. Astrocytoma and ependymoma yield similar predictions. Both tumor types have overlapping characteristics (Kim et al., 2014): high intensity signals on T2w, comparable enhancement patterns, similar size (astrocytoma: 2-19 vertebral bodies, ependymoma: 2-13 vertebral bodies (Baleriaux, 1999)), etc. Predictions with hemangioblastoma as input diverge from the other tumor types. Hemangioblastoma predictions reflect their characteristics: small tumor cores intensely enhanced in Gd-e T1w, as seen in Figure 4. When inputting the hemangioblastoma label for the astrocytoma (first row of Figure 5) no prediction is given since the Gd-e T1w modality has moderate enhancement. Similarly, for the ependymoma, only the most Gd-enhanced portion of the tumor is predicted when assigning the hemangioblastoma label with FiLM (second row of Figure 5). The results from Table 3 and Figure 4 - 5 confirm that FiLM layers are able to learn characteristics from the metadata that are relevant for the segmentation.

