# OpenReview forum: "Benefits of Linear Conditioning for Segmentation using Metadata"
_MIDL.io/2021/Conference — MIDL 2021_

### Official Review · AnonReviewer2 · 2021-03-05

**Confidence:** 4
**Preliminary Rating:** 2
**Final Rating:** 3

**Summary:**

The paper proposes to use linear conditioning, in the form of FiLM, to improve the performance of U-net segmentation tasks. The paper claims the conditioning can address problems in few-shot learning, missing segmentations, and adaptation. Empirical results on tumor segmentation show tumor type as conditional input could improve the Dice score. Empirical results on organ segmentation show that the proposed conditioning improves organ segmentation in the few-shot regime.

**Strengths:**

1. The use of meta-data as input for conditioning in segmentation tasks is novel and interesting.
2. The application of the proposed approach to few-shot segmentation is an important area of research.

**Weaknesses:**

In general, I think the writing of this paper severely lacks clarity and detail regarding the training and evaluation procedures.

1. In section 2.2.2 regarding evaluation, why both train/valid/test split and cross-validation are used? How are the data shuffled exactly in cross-validation?

2. In section 2.3.2., for the few-shot experiments, what are the relationships between the first two paragraphs? Are the models trained sequentially (one pretrains another) or independently. Why the model is first trained on the whole dataset, then evaluated against few-shot learning on the same classes? Does it not defeat the purpose of few-shot learning because the model has observed more data before?

3. In section 2.3.2, the data split for evaluation is quite unusual and the authors should explain why always keep one class 12 and vary the other. Why not do few-shot training and evaluation on both tasks?

4. In section 2.3.2, "For a model trained on 2 kidney subjects and 12 spleen subjects, the model would be tested on 25 spleen subjects not included in the training or validation set." Is this a typo, should be evaluated on kidney rather than spleen?

5. In section 2.3.2, "The dataset was split per patient with the following proportions: 60% training, 20% validation, 20% testing." If I understand correctly, why not train and evaluate on different patients for better generalization estimations?

6. In section 2.2.2, "The first scenario used the FiLM architecture without any input metadata." What does it mean "without input" since FiLM must require some kind of input? What kind of input is used in this case?

**Deanonymize Review:**

no

**Final Rating Justification:**

The authors addressed my concerns about this paper and I revise my score to "Weak Accept."



**Justification Of The Preliminary Rating:**

I like the idea of using linear conditioning for multi-task/class segmentation, but the writing of this paper lacks sufficient clarity especially regarding the training and evaluation procedures. The evaluation setup for few-shot experiments is unusual and lacks justification.

**Paper Type:**

both

**Questions To Address In The Rebuttal:**

Please see above.

**Special Issue:**

no

---

> ### Author Response · Authors · 2021-03-18
> **Author response**
>
> > 1. In section 2.2.2 regarding evaluation, why both train/valid/test split and cross-validation are used? How are the data shuffled exactly in cross-validation?
>
> Thank you for pointing this out. We misused “cross-validation”. We should have mentioned “random splits” rather than “cross-validations”. The dataset is randomly shuffled to obtain 60% of the subjects in the training set, 20% in the test set, and 20% in the validation set. This has been corrected in the revised version of the paper.
>
> > 2. In section 2.3.2., for the few-shot experiments, what are the relationships between the first two paragraphs? Are the models trained sequentially (one pretrains another) or independently. Why the model is first trained on the whole dataset, then evaluated against few-shot learning on the same classes? Does it not defeat the purpose of few-shot learning because the model has observed more data before?
>
> Yes, this would definitely defeat the purpose of few-shot learning. The models trained on the whole dataset (paragraph 1) and the models trained with only a few subjects for different dataset sizes (paragraph 2) were trained independently. We rephrased this passage to clarify the independence of training of both sub-experiences:
>
> *“Second, the performance on small and unbalanced datasets was assessed with an independent sub-experiment: FiLMed U-Net was trained on subdatasets of the spleen and kidney datasets.”*
>
> > 3. In section 2.3.2, the data split for evaluation is quite unusual and the authors should explain why always keep one class 12 and vary the other. Why not do few-shot training and evaluation on both tasks?
>
> This experimental design was chosen to test unbalanced datasets (e.g., 2 kidneys with 12 spleens) and performance with limited amounts of data.  This simulates a situation where one has an annotated dataset of kidneys for instance and would like to have a model predicting both kidney and spleen. It illustrates how many annotations of the spleen are required to reach decent performances. 12 subjects were chosen as a compromise between reaching decent segmentation while having few images.
>
> > 4. In section 2.3.2, "For a model trained on 2 kidney subjects and 12 spleen subjects, the model would be tested on 25 spleen subjects not included in the training or validation set." Is this a typo, should be evaluated on kidney rather than spleen?
> Yes, it is indeed a typo. Thank you for pointing it out, it has now been corrected:
>
> *“For a model trained on 2 kidney subjects and 12 spleen subjects, the model would be tested on 25 kidney subjects not included in the training or validation set.”*
>
> > 5. In section 2.3.2, "The dataset was split per patient with the following proportions: 60% training, 20% validation, 20% testing." If I understand correctly, why not train and evaluate on different patients for better generalization estimations?
>
> Yes, absolutely this is what we did. 20% of subjects were used for evaluation and were never presented to the model during training or validation. I think the confusion might come from the “per patient”. We rephrased it to make this clearer:
>
>  *"The subjects were split with the following proportions: 60% training, 20% validation, 20% testing."*
>
> >  6. In section 2.2.2, "The first scenario used the FiLM architecture without any input metadata." What does it mean "without input" since FiLM must require some kind of input? What kind of input is used in this case?
>
> Great question! You are absolutely right, FiLM always requires input data. To simulate the absence of data, we always included the same input vector. This way no valuable information is passed to the model, but the architecture stays exactly the same. We have now clarified this in the manuscript:
>
> *“To simulate the absence of metadata, always the same input vector was passed through FiLM, hence informative data is seen by the model.”*

---

> > ### Comment · AnonReviewer2 · 2021-03-18
> > **Thanks for the response**
> >
> > The authors addressed my concerns about this paper and I revise my score to "Weak Accept."

---

### Official Review · AnonReviewer1 · 2021-03-05

**Confidence:** 5
**Preliminary Rating:** 2
**Final Rating:** 3

**Summary:**

The main contribution of this work is the use of a feature linear conditioning from text labels to improve the segmentation performance when sufficient number of labeled image sets are unavailable. The authors used two different datasets, a large spinal cord dataset and a relatively modest sized abdominal organs dataset for their analysis.

**Strengths:**

The idea of incorporating known characteristics like tumor type, location, etc are interesting to capture to improve the performance a deep learning method. Although the authors do not present it this way, the second case of providing only which organs are present but not the segmentation for subset of cases could also be cast as a weak learning. The results though for the second case are not better than the published results, could still be considered potentially interesting (if its explained a little bit better).

**Weaknesses:**

I think the way performance comparisons were done is not fair for the second problem.  It would help if the authors explained why the experiment was performed by training a separate Unet for each organ? The more standard approach would be to train a Unet with all the labels as one network instead of separate network. In order to show that there is improvement to be made using the proposed approach, one would expect to see comparison against a standard way in which training will be done.

The methods could be explained a little better.



**Deanonymize Review:**

no

**Detailed Comments:**

The main concern is whether the experiments were done in a way to ensure fair comparisons. The tests for the first problem seem reasonable and the performance improvement is quite clear. But its hard to know if the results are indeed improved for the second scenario. Especially given the small number of training/testing cases.

The experiments with 2, 4, 6, 8, training cases could be explained better and is rather confusing to follow.  Given such a small number of cases, its not clear how reasonable generalization is achieved with a fairly large batch size of 8.

Also, it is not clear what the text attributes are. For the second scenario it looks like its just the names of the organs, and for the first dataset is it just a one tumor type or does it include other attributes as well. Regardless, one issue is that this is fairly low dimensional and this is being injected in every layer. Given that, how much does it help to do the feature conditioning in every layer. It would be nice to see some ablation experiments in this regard and more details on exactly what the attributes were.

**Final Rating Justification:**

Authors have addressed my comments and concerns.

**Justification Of The Preliminary Rating:**

A lot of details in the method are left unexplained and could be explained better. Also the setting of the second experiment seems somewhat questionable. Also the performance doesn't necessarily get better for the second approach even in the setting used for the comparison, which uses separately trained networks.

**Paper Type:**

methodological development

**Questions To Address In The Rebuttal:**

Please provide rationale and clearly explain why comparison for 2DUnet without FiLM was done as a separate network trained for each organ?

Also, the point of experiments (at least how they are set up) for the 2,4,6,8 examples is unclear. Please provide evidence that the network does converge. These seem like very small number of data to obtain any meaningful generalization.

Please explain clearly what the text attributes/dimensionality of the embedding used etc is. Without sufficient details its hard to understand why this even works.

**Special Issue:**

no

---

> ### Author Response · Authors · 2021-03-18
> **Author response**
>
> > Please provide rationale and clearly explain why comparison for 2DUnet without FiLM was done as a separate network trained for each organ?
>
> This comment is addressed in the response addressed to all reviewers.
>
>
> > The methods could be explained a little better.
>
> Thank you for this comment. We clarified the methodology according to your comments and added a figure in the appendix to have a visual representation of the experimental design of experiment 2.
>
> *“ The experimental design of this sub-experiment is presented in appendix A.”*
>
> > Also, the point of experiments (at least how they are set up) for the 2,4,6,8 examples is unclear. Please provide evidence that the network does converge. These seem like very small number of data to obtain any meaningful generalization.
>
> For the few-shot experiment, we wanted to simulate small and unbalanced datasets. To do so, each FiLM experiment was composed of the following:
>
> 2 kidneys  - 12 spleens --> testing on kidneys only
>
> 4 kidneys  - 12 spleens --> testing on kidneys only
>
> 6 kidneys  - 12 spleens --> testing on kidneys only
>
> 8 kidneys  - 12 spleens --> testing on kidneys only
>
> 12 kidneys - 12 spleens --> testing on kidneys only
>
> 12 kidneys - 12 spleens --> testing on spleen only
>
> 12 kidneys  - 8 spleens  --> testing on spleen only
>
> 12 kidneys  - 6 spleens --> testing on spleen only
>
> 12 kidneys  - 4 spleens --> testing on spleen only
>
> 12 kidneys  - 2 spleens --> testing on spleen only
>
>
> One iteration of the experiment corresponds to 10 FiLM models, and 10 U-Net models trained only on 2, 4, 6, 8, 12 spleen or kidneys, respectively. The reason why the regular U-Nets were only trained on one organ at the time is that the missing labels would severely hamper the performance (as described in the comment addressed to all reviewers).
>
> To ensure convergence, the models were trained until the validation loss plateaued for 50 epochs (early stopping with epsilon = 0.001). We also repeated each experiment 10 times, which totals 200 trainings (100 with FiLM and 100 with U-Net) to ensure we could draw conclusions from the data and make a statistical analysis.
>
> > Please explain clearly what the text attributes/dimensionality of the embedding used etc is. Without sufficient details its hard to understand why this even works.
>
> As you mentioned in your review, the attributes input into FiLM were the tumor type and organ name were used for experiments 1 and 2 respectively. The labels were one-hot-encoded. For instance, since there were 3 tumor types, the vector [1, 0, 0] represented astrocytomas, [0, 1, 0] ependymomas, and [0, 0, 1] hemangioblastomas. The one-hot-encoded metadata is then passed into the FiLM generator (i.e., MLP) and generates one value of γ and β for each feature map. γ multiplies the convolutional feature maps while β is added. Hence, this leads to a modulated feature map that in the end modifies the final output.
> In other words, the FiLM approach generates a set of affine transformations (one transformation per feature map) and this set changes based on the input vector (e.g., tumor or organ type).
>
> Indeed, the question about the specific benefits of these low dimensional modulations is extremely relevant. In order to assess the specific benefits of this FILM modulation, we performed an experiment (experiment 1), where the input vector is always the same (e.i., instead of changing it based on the tumor or organ type, we only set it to “[1,0,0]”)-- results are presented in Table 1. This isolates the impact of the metadata on the training and shows that the model was able to learn from the metadata provided.
>
> > Also the performance doesn't necessarily get better for the second approach
>
> Yes indeed, this could in fact have extremely relevant applications. In the second experiment trained on the whole data, we wanted to demonstrate that a FiLMed multi-class model trained with missing labels could reach optimal Dice scores, which is not possible with a regular multi-class U-Net. Most available datasets have only one class segmented, but one might need to segment multiple classes (e.g., organ segmentation). Instead of creating a model for each class, here, we can generate a single model that is easier to package and use in routine.

---

### Official Review · AnonReviewer4 · 2021-03-07

**Confidence:** 5
**Preliminary Rating:** 3
**Recommendation:** Poster

**Summary:**

This paper proposes a framework for incorporating metadata into a Unet by training an MLP (with shared weights across layers) to propose layer specific affine scaling parameter. The method is tested on a multi-organ segmentation problem and for tumour segmentations with tumour class as a meta label.


**Strengths:**

This paper builds on previous work on using FiLM to incorporate meta-data into networks. This is a difficult problem when training sets are small and it definitely looks to be working for the tumour problem specifically. The validation experiments are fairly thorough and do indicate the approach might also offer some advantages for multi-label segmentation of data sets with few examples. Overall the paper is very clearly motivated, the methods are well explained, figures are clear and the results are transparently presented.



**Weaknesses:**

Much of the validation is reserved for the discussion of the multi-organ segmentation task but the authors do not compare against a single U-net trained to detect all organs. It seems the method has more merit for problems with confound variables,

The analysis of the impact of data set size is welcome but it might have been more impactful for the tumour classification problem and for  greater range of N


**Deanonymize Review:**

no

**Justification Of The Preliminary Rating:**

This paper is well motivated, clearly written, well presented and validation is pretty convincing. In my opinion, the paper would have been more convincing if more space had been dedicated to more evaluations of the spinal tumour problem or evaluation  on more datasets where segmentation problem is clearly confounded by nuisance variables

**Paper Type:**

methodological development

**Questions To Address In The Rebuttal:**

1. Comparison of the multi-organ classification against a single multi class u-net
2. More evaluation of the tumour results perhaps including impact of dataset size

**Special Issue:**

no

---

> ### Author Response · Authors · 2021-03-18
> **Author response**
>
> > Much of the validation is reserved for the discussion of the multi-organ segmentation task but the authors do not compare against a single U-net trained to detect all organs. It seems the method has more merit for problems with confound variables,
>
> This comment is addressed in the response addressed to all reviewers.
>
>
> > The analysis of the impact of data set size is welcome but it might have been more impactful for the tumour classification problem and for greater range of N
>
> Yes, we agree it would have been interesting to analyze the impact on dataset size for the experiment on the tumor segmentation as well. This experiment on organ segmentation (Figure 2) regrouped 200 models trained for approximately 100 epochs each. Unfortunately, due to our limited access to GPU clusters (this experiment would have taken ~20 days to run on the 4 GPU cards we have access to), we could not include this experiment for the tumor dataset. The purpose of Figure 2 was to get an intuition of how FiLM could help in a low label regime. The generalization of this trend should be investigated further on other tasks with different balancement, data quality, data size, number of output classes, etc.

---

### Official Review · AnonReviewer3 · 2021-03-07

**Confidence:** 3
**Preliminary Rating:** 3
**Recommendation:** Oral, Poster
**Final Rating:** 3

**Summary:**

The paper proposes to integrate FiLM (Feature-wise Linear Modulation) layer with U-Net to unable use prior information (Ex. Tumour sub-type, segmentation task information). The experiments and results show that by integrating prior information in the model, better segmentation performance can be achieved. Experiments are well motivated with a clear objective.

**Strengths:**

* The paper proposes a novel method to combine metadata into a segmentation model. As per my knowledge, this is the first paper to achieve this information.
* Experiment-1 shows a clear advantage of including prior information using FiLM layers.
* Experiment-2 shows that the inclusion of prior information plays a bigger role when limited annotations are available.
* Inclusion of the statistical significance test is a big plus. This is in addition to publicly available code.
* Discussion section explores the possibility of future work in a real clinical scenario. This should serve as a good motivation for future papers.


**Weaknesses:**

* The paper misses a paper that uses FiLM layers [1]. In this paper, the FiLM layer is shown to help in situations when limited annotations are available. This work is closest to experiment-2 in the proposed paper. Discussion regarding this is necessary.
* Similarly, the paper also misses a reference to [2] which uses conditioning on Tumour type. This should be included in the context of Experiment-1.
* In experiment-2, only multi-class FiLMed U-Net and single-class regular U-Net are compared. It was expected to also compare multi-class U-Net without FiLM layer, as it would allow disentangling the effect of FiLM layers from just having more annotated data (transfer learning).

[1] Chartsias, A., Papanastasiou, G., Wang, C., Semple, S., Newby, D.E., Dharmakumar, R. and Tsaftaris, S.A., 2020. Disentangle, align and fuse for multimodal and semi-supervised image segmentation. IEEE transactions on medical imaging.

[2] Rebsamen, M., Knecht, U., Reyes, M., Wiest, R., Meier, R. and McKinley, R., 2019. Divide and conquer: stratifying training data by tumor grade improves deep learning-based brain tumor segmentation. Frontiers in neuroscience, 13, p.1182.

**Deanonymize Review:**

no

**Detailed Comments:**

* Experiment on a full Medical Segmentation Decathlon dataset can serve as a good benchmark for future work.
* There is a typo in the paper. Sec:2.3.2 "The models were tested on 25 subjects of the class with the least subject. For a model trained on 2 kidney subjects and 12 spleen subjects, the model would be tested on 25 spleen subjects not included in the training or validation set." Here both sentences are contradicting.

**Final Rating Justification:**

I would like to keep my original rating. The authors have written a good rebuttal but still miss some of the points, which doesn't allow me to improve the rating of the paper.

**Justification Of The Preliminary Rating:**

The paper proposes a novel method to combine prior information. Though it is missing some recent references, experiments and results show the advantage of the proposed method. I particularly liked the discussion section which clearly motivates future work.

**Paper Type:**

methodological development

**Questions To Address In The Rebuttal:**

* Mainly, all points of the weaknesses section.

**Special Issue:**

yes

---

> ### Author Response · Authors · 2021-03-18
> **Author reponse**
>
> > The paper misses a paper that uses FiLM layers [1]. In this paper, the FiLM layer is shown to help in situations when limited annotations are available. This work is closest to experiment-2 in the proposed paper. Discussion regarding this is necessary.
>
> > Similarly, the paper also misses a reference to [2] which uses conditioning on Tumour type. This should be included in the context of Experiment-1.
>
> Thank you for suggesting adding these highly relevant references. They have now been incorporated into the revised version of the paper:
>
> *“For instance, knowledge of the tumor type could provide useful information to the model. Rebsamen et al. demonstrated that by stratifying the learning by brain tumor type, high-grade glioma, or low-grade glioma, segmentation could be improved. With FiLM, the tumor type information can be included without requiring multiple models as done in Rebsamen et al.”*
>
> *“In the medical image field, FiLM was leveraged for learning when limited or no annotation is available for one modality Chartsias et al. Image reconstruction was performed with FiLM to enable self-supervised learning of the anatomical and modality factors of an image. Modality factors (e.g., encoding pixel intensities) were passed through FiLM to modulate anatomical factors (i.e., encoding semantic information) generating a reconstructed image of a given modality. While in Chartsias et al. information extracted from the image is used for modulation, in this work, we want to assess the impact of integrating metadata which is not directly encoded in the image. ”*
>
>
> > In experiment-2, only multi-class FiLMed U-Net and single-class regular U-Net are compared. It was expected to also compare multi-class U-Net without FiLM layer, as it would allow disentangling the effect of FiLM layers from just having more annotated data (transfer learning).
>
> This comment is addressed in the response addressed to all reviewers.
>
> > Experiment on a full Medical Segmentation Decathlon dataset can serve as a good benchmark for future work.
>
> Yes, absolutely! This is definitely an avenue we would like to explore in future work. Thank you for this excellent suggestion.
>
>
> > There is a typo in the paper. Sec:2.3.2 "The models were tested on 25 subjects of the class with the least subject. For a model trained on 2 kidney subjects and 12 spleen subjects, the model would be tested on 25 spleen subjects not included in the training or validation set." Here both sentences are contradicting
>
> Thank you for pointing this out, indeed it should be 25 **kidney** subjects. This has now been corrected in the updated version of the manuscript:
>
> *“For a model trained on 2 kidney subjects and 12 spleen subjects, the model would be tested on 25 kidney subjects not included in the training or validation set.”*

---

> > ### Comment · AnonReviewer3 · 2021-03-21
> > **Response to the rebuttal**
> >
> > Thank you for incorporating the related work and correcting the typos.
> >
> > I still do believe that a better comparison of multi-class U-Net without FiLM layer can be done in a better way against a publicly available decathlon method [1][2].
> >
> > [1] Isensee, F., Petersen, J., Klein, A., Zimmerer, D., Jaeger, P.F., Kohl, S., Wasserthal, J., Koehler, G., Norajitra, T., Wirkert, S. and Maier-Hein, K.H., 2018. nnu-net: Self-adapting framework for u-net-based medical image segmentation. arXiv preprint arXiv:1809.10486.
> > [2] https://github.com/MIC-DKFZ/nnUNet

---

### Author Response · Authors · 2021-03-18
**General comment addressed to all reviewers**

We would like to thank all the reviewers for their insightful feedback and detailed comments. A revised version of the manuscript is included in the rebuttal.

> In experiment-2, only multi-class FiLMed U-Net and single-class regular U-Net are compared. It was expected to also compare multi-class U-Net without FiLM layer, as it would allow disentangling the effect of FiLM layers from just having more annotated data (transfer learning).

Excellent point raised by all the reviewers. When training a multi-class segmentation model, all the classes need to be segmented on each image, otherwise, missing labels will hamper the learning. In our case, the chosen datasets only have one segmented organ per subject, hence it was not possible to directly test multi-class without implementing a custom strategy to deal with the missing annotations. The “ground truth” only contains one organ out of three, thus the model ends up segmenting only part of each organ. Despite the above-mentioned rationale and to further investigate the reviewers’ suggestion, we did try to train a multi-class U-Net without FiLM model, and as expected the mean Dice score was  41.7 $\pm$ 16.0% (averaged Dice across all tasks). Moreover, this poor metric further demonstrates the relevance of FiLM for training with missing labels. Using separate U-Nets was the most straightforward way to reach high performances for the baseline with the data available. We agree this information should be stated explicitly in our results section to clarify this experimental design choice, and the information is now included in Table 2 and in the results section:

*“As a reference, a multi-class 2D U-net was trained with this same dataset which has missing labels (i.e., only one organ is labeled per scan). Poor performance was reached with an average Dice score of 41.7 $\pm$ 16.0 for all classes combined as only part of each organ is segmented. This result illustrates the hindered learning caused by the missing annotations.”*

Another avenue we considered for achieving a fair assessment of the specific benefits of FiLM was to use transfer learning. In this scenario, we would train on class #1 and fine-tune on class #2 to make predictions on class #2. However this is a two-step process, so it would not really be comparable to FiLMed U-Net which doesn’t require two steps. Also, the transfer learning approach becomes more complicated when we consider more than two classes (e.g., multi-class model for liver, spleen, and kidneys).

---

### Meta-Review · Area_Chair1 · 2021-03-29

**Recommendation:** Accept (Poster)

**Metareview:**

All reviewers agree that the idea of this paper is novel and interesting to MIDL community. While there were major questions on several details of the proposed model and experimental evaluation, the authors have made to address them appropriately. Based on the final ratings of all reviewers, I am happy to recommend acceptance of the current manuscript.

**Paper Type:**

methodological development

---

### Decision · Program_Chairs · 2021-03-31

Accept